# Off-Clamp Robotic-Assisted Partial Nephrectomy: Retrospective Comparative Analysis from a Large Italian Multicentric Series

**DOI:** 10.3390/cancers17162645

**Published:** 2025-08-13

**Authors:** Angelo Porreca, Filippo Marino, Davide De Marchi, Marco Giampaoli, Francesca Simonetti, Antonio Amodeo, Paolo Corsi, Francesco Claps, Daniele Romagnoli, Alessandro Crestani, Luca Di Gianfrancesco

**Affiliations:** 1Department of Urology, Humanitas Gavazzeni, 24125 Bergamo, Italy; angelo.porreca@hunimed.eu (A.P.); davide.demarci@gavazzeni.it (D.D.M.); marco.giampaoli@gavazzeni.it (M.G.); francesca.simonetti@gavazzeni.it (F.S.); luca.digianfrancesco@gavazzeni.it (L.D.G.); 2Humanitas University, 20089 Rozzano, Italy; 3Department of Urology, Veneto Institute of Oncology—IOV IRCCS, 35128 Padua, Italy; antonio.amodeo@iov.veneto.it (A.A.); paolo.corsi@iov.veneto.it (P.C.); francesco.claps@iov.veneto.it (F.C.); 4Department of Urology, Policlinico Abano Terme, 35031 Abano Terme, Italy; infodottromagnoli@gmail.com; 5Department of Urology, Azienda Sanitaria Universitaria Friuli Centrale, 33100 Udine, Italy; alessandro.crest@gmail.com

**Keywords:** robotic-assisted partial nephrectomy, off-clamp, on-clamp, renal function preservation, oncologic control, robotic nephrectomy outcomes, ischemia-free nephrectomy

## Abstract

**Simple Summary:**

Kidney tumors are often treated with surgery that removes only the tumor while preserving as much healthy kidney tissue as possible. Traditionally, surgeons temporarily stop blood flow to the kidney during surgery to reduce bleeding, but this can damage kidney function. In this study, we looked at an alternative method called off-clamp robotic-assisted partial nephrectomy, where the tumor is removed without stopping the kidney’s blood flow. We analyzed data from over 500 patients treated at multiple hospitals and compared their outcomes to patients who had the standard procedure with blood flow clamped. We found that patients undergoing the off-clamp technique had shorter surgeries, less blood loss, and fewer complications. Importantly, their kidney function remained stable, and cancer control was excellent, with very few cases of cancer recurrence. Most complications were mild, and no major kidney injuries occurred. These results suggest that the off-clamp method is safe, preserves kidney health, and effectively treats cancer. This technique could benefit patients, especially those at risk of future kidney problems. However, more long-term studies are needed to confirm these results and help doctors decide which patients are best suited for this approach.

**Abstract:**

Objective: To evaluate the perioperative outcomes, functional impact, and oncologic efficacy of off-clamp robotic-assisted partial nephrectomy (RAPN) in patients with renal masses across multiple high-volume centers. Materials and Methods: We conducted a retrospective multicenter study including 563 patients (group 1) who underwent clampless RAPN between January 2018 and December 2024. Patients with solitary kidneys, tumors >7 cm, or prior renal surgery were excluded. The standardized surgical technique involved tumor resection without clamping of the renal artery, followed by the use of hemostatic agents and standard/selective suturing of the resection bed on demand. Patients in group 1 were compared to 244 consecutive patients treated in the same centres and treated with RAPN with an on-clamp procedure (group 2). Primary outcomes included operative time, blood loss, and complications, while secondary outcomes assessed renal function preservation and oncologic control at an at least 12-month follow-up. Results: The median operative time was 118 min (IQR: 100–140 min), and median estimated blood loss was 150 mL (range: 50–400 mL). The overall complication rate was 9.2%, with most classified as Clavien–Dindo Grade I–II. No intraoperative conversions to open surgery were recorded. Renal function was well preserved, with a median estimated glomerular filtration rate (eGFR) decline of 4.1% at three months (*p* > 0.05), and no cases of acute kidney injury. Oncologic outcomes were favorable, with a positive surgical margin rate (PSM) of 2.4% and two cases of tumor recurrences (0.36%) documented at a 12-month follow-up. Conclusions: The off-clamp RAPN is a safe and effective nephron-sparing approach, offering significant renal function preservation while maintaining oncologic efficacy. This technique minimizes ischemia–reperfusion injury and post-surgical fibrosis, providing a viable alternative to on-clamp RAPN. Further prospective trials are warranted to confirm long-term benefits and refine patient selection criteria.

## 1. Introduction

Renal cell carcinoma (RCC) is the most prevalent form of kidney cancer, representing approximately 90% of all renal malignancies [1]. Globally, RCC accounts for around 2–3% of all adult malignancies and is associated with significant morbidity and mortality [2]. The widespread use of abdominal imaging modalities, such as ultrasonography, computed tomography (CT), and magnetic resonance imaging (MRI), led to an increased detection rate of small renal masses (SRMs), defined as renal tumors less than or equal to 4 cm in maximum diameter. Many of these tumors are detected incidentally during imaging performed for unrelated conditions [3]. This phenomenon contributed to a stage migration in RCC, with a greater proportion of patients presenting with localized, smaller tumors amenable to nephron-sparing treatments [4]. Partial nephrectomy (PN) has become the standard of care for clinical T1 renal masses and is strongly recommended by international guidelines, including the European Association of Urology (EAU) and the American Urological Association (AUA) [5,6]. Compared to radical nephrectomy (RN), PN offers equivalent oncologic control while preserving renal function, thereby reducing the long-term risks of chronic kidney disease (CKD), cardiovascular events, and all-cause mortality [7]. The paradigm shift toward nephron-sparing surgery is especially critical given the increasing prevalence of CKD and comorbidities such as hypertension and diabetes in the aging population [8]. The emergence of robotic-assisted partial nephrectomy (RAPN) significantly influenced the surgical management of renal tumors. Since its introduction, RAPN has shown improved perioperative outcomes compared to open and pure laparoscopic approaches, including reduced blood loss, shorter hospital stays, and faster recovery [9,10]. The robotic platform provides high-definition 3D visualization, enhanced dexterity, tremor filtration, and ergonomic advantages that enable precise dissection and reconstruction, especially in challenging tumor locations [11]. Consequently, RAPN has been widely adopted in high-volume centers as a preferred technique for performing PN. Traditionally, PN is performed under temporary clamping of the renal artery—and occasionally the renal vein—to minimize intraoperative bleeding and enhance visualization during tumor excision. However, clamping introduces a period of warm ischemia, which can lead to ischemia–reperfusion injury and irreversible damage to nephrons, particularly in patients with preexisting renal impairment or solitary kidneys [12,13]. Studies demonstrated that even short durations of warm ischemia can result in long-term renal function deterioration, underscoring the need for strategies that minimize or eliminate ischemic insult [14,15]. In response to this challenge, alternative surgical strategies have been proposed to reduce or avoid warm ischemia. These include early unclamping, segmental or selective arterial clamping, zero ischemia (clampless) techniques, and preoperative super-selective embolization [16]. Among these, clampless PN—also known as off-clamp or zero-ischemia PN—entails tumor resection without vascular clamping, thereby preserving continuous perfusion to the kidney throughout the procedure [17]. This technique necessitates meticulous surgical planning, refined dissection skills, and the use of advanced hemostatic agents to control bleeding in the absence of vascular occlusion. Initial experiences with clampless PN, both laparoscopic and robotic, demonstrated its potential to preserve renal function more effectively than standard on-clamp PN, especially in patients with baseline renal insufficiency [18,19]. However, clampless PN is technically more demanding and may be associated with increased intraoperative blood loss and a steeper learning curve. While several single-institution series reported encouraging outcomes, the generalizability and reproducibility of these findings in broader clinical settings remain uncertain. Additionally, concerns persist regarding the oncologic safety of clampless PN, particularly in larger or more complex tumors, where the risk of positive margins or recurrence could be higher without the hemostatic benefit of clamping. The anatomical complexity of renal tumors, commonly quantified using scoring systems such as the RENAL nephrometry score, further influences the feasibility and safety of clampless techniques. Tumors located near the renal hilum or with endophytic growth patterns are often considered more technically challenging to resect without vascular control. Consequently, careful patient selection, preoperative imaging assessment, and surgical expertise are paramount when considering clampless approaches. Despite the growing interest in clampless RAPN, the current literature remains limited by small sample sizes, heterogeneity in surgical technique, and variable definitions of functional and oncologic endpoints. There is a need for robust, multicenter data to validate the efficacy, safety, and reproducibility of clampless RAPN, particularly when standardized protocols are followed across high-volume institutions. The present study aims to address this knowledge gap by evaluating the perioperative, functional, and oncologic outcomes of clampless RAPN across three high-volume urological centers in Italy. By analyzing a large cohort of patients undergoing standardized clampless RAPN procedures, we sought to provide comprehensive evidence supporting the feasibility, safety, and effectiveness of this technique in the management of localized T1 renal tumors. Through this collaborative effort, we also aimed to identify factors predictive of successful clampless surgery and contribute to the refinement of patient selection criteria and surgical planning. This investigation not only provides insights into the evolving role of clampless RAPN, but also reflects broader trends in precision surgery, where individualized, nephron-sparing approaches are prioritized to optimize long-term outcomes in patients with renal tumors.

## 2. Materials and Methods

### 2.1. Study Design and Setting

This retrospective multicenter cohort study was conducted at three high-volume tertiary urological centers in Italy: Humanitas Gavazzeni (Bergamo), Veneto Institute of Oncology (IOV)—IRCCS (Padua), and Policlinico Abano Terme (Abano Terme). The study protocol adhered to the Declaration of Helsinki and received ethical approval from the institutional review boards of all participating centers. Informed consent was obtained from all patients prior to surgery, including consent for data collection and follow-up.

### 2.2. Objective

To evaluate the perioperative outcomes, functional impact, and oncologic efficacy of off-clamp RAPN compared to on-clamp RAPN in patients with renal masses across multiple high-volume centers.

### 2.3. Patient Selection Criteria

We included all adult patients (≥18 years) who underwent elective clampless robot-assisted partial nephrectomy (RAPN) for clinically localized renal masses between January 2018 and December 2024. Eligible tumors were solitary, with a maximum diameter ≤7 cm, corresponding to clinical stage T1a or T1b renal cell carcinoma (RCC), as confirmed by preoperative imaging.

Exclusion criteria were as follows:Multifocal tumors or bilateral renal masses,Patients with solitary kidney or previous ipsilateral renal surgery,Conversion to hilar clamping or open surgery intraoperatively,Tumors >7 cm or locally advanced disease (≥T2),Lack of complete perioperative or follow-up data.

### 2.4. Preoperative Assessment

Preoperative workup included a comprehensive clinical evaluation, laboratory tests including serum creatinine and estimated glomerular filtration rate (eGFR) calculation using the CKD-EPI formula, and cross-sectional imaging with contrast-enhanced computed tomography (CT) or magnetic resonance imaging (MRI) to characterize tumor size, location, and complexity.

Tumor anatomical complexity was assessed using the RENAL nephrometry scoring system, which accounts for tumor radius (size), exophytic/endophytic properties, nearness to collecting system or sinus, anterior/posterior descriptor, and location relative to polar lines [14]. Scores were stratified as low (4–6), moderate (7–9), or high (≥10) complexity.

### 2.5. Surgical Technique

All RAPN procedures were performed using the Da Vinci robotic platforms (Si or Xi, Intuitive Surgical, Sunnyvale, CA, USA). The decision for a transperitoneal or retroperitoneal approach was individualized based on tumor location, surgeon experience, and patient habitus. Pneumoperitoneum was established using the AirSeal^®^ system (CONMED, Utica, NY, USA), maintaining a consistent pressure of 12–15 mmHg.

The renal hilum was meticulously dissected and isolated at the outset of surgery to allow rapid vascular control if bleeding occurred; however, clamping was avoided to maintain the clampless approach. After tumor localization, intraoperative ultrasound was routinely employed to delineate tumor margins and assess depth.

Tumor excision was performed primarily by cold scissors or monopolar curved scissors, with bipolar energy used sparingly via ProGrasp forceps for precise hemostasis. Enucleation or enucleoresection techniques were chosen according to tumor characteristics: pure enucleation for well-circumscribed tumors, and enucleoresection when a margin of normal parenchyma was warranted for oncologic safety.

Hemostasis was achieved by meticulous identification and selective ligation of visible arterial or venous branches using 3-0 or 4-0 absorbable sutures (Vicryl^®^ or poliglecaprone). Hemostatic agents, including oxidized regenerated cellulose (Surgicel^®^), gelatin matrix (Floseal^®^), and fibrin glue (Tisseel^®^), were routinely applied to the resection bed to promote coagulation and prevent oozing.

Renorrhaphy was selectively performed based on intraoperative assessment of the depth and size of the parenchymal defect and collecting system entry. When required, a single-layer closure of the renal parenchyma was performed using barbed suture (3-0 V-Loc™, Medtronic, Dublin, Ireland) to approximate tissue without ischemic insult. The collecting system, if opened, was repaired with running 4-0 absorbable sutures.

A closed-suction drain was placed in the renal fossa at the conclusion of the procedure in all cases to monitor for bleeding or urinary leaks.

With particular attention to highly complex cases, the tumors were managed using a robotic off-clamp technique with a focus on minimal manipulation of hilar vessels and selective suturing based on intraoperative bleeding. In these higher-complexity cases, the following hemostatic principles were consistently applied: meticulous dissection and enucleation of the tumor were performed under continuous perfusion using cold scissors, with real-time intraoperative ultrasound guiding resection planes and identifying proximity to major vessels or the collecting system; bleeding control was achieved primarily through careful surgical technique: gentle traction, minimal parenchymal trauma, and staged tumor mobilization helped limit diffuse bleeding without requiring hilar clamping; topical hemostatic agents (e.g., oxidized regenerated cellulose or fibrin-based sealants) were applied immediately after tumor excision. These agents were used not only to tamponade oozing surfaces, but also to promote coagulation in the tumor bed; selective parenchymal suturing was employed only if there was persistent focal bleeding after compression and topical measures. In such cases, targeted figure-of-eight sutures (using 3-0 or 2-0 absorbable monofilament or barbed sutures) were placed in a tension-free manner to minimize tissue distortion and unnecessary ischemic damage; routine renorrhaphy was intentionally avoided, even in high-complexity cases, to preserve as much functional parenchyma as possible and to stay consistent with the principles of zero ischemia; in instances where tumor base bleeding obscured visualization despite the above measures, temporary manual compression with robotic instruments or laparoscopic gauze was applied for 2–3 min before reassessment. This conservative and individualized approach to hemostasis was effective in maintaining a low intraoperative complication rate without resorting to global hilar clamping or extensive renorrhaphy. In our series, no cases required conversion to hilar clamping, and we can confirm that no transient or aborted clamping was initiated or proposed during any of the procedures. In all cases, a strict intent-to-treat off-clamp strategy was followed from the outset. The decision to proceed without clamping was confirmed intraoperatively after evaluating tumor exposure, bleeding control during initial resection, and accessibility of the lesion with robotic instruments. Surgeons were instructed to document any intraoperative scenario in which clamping was considered, even if ultimately avoided; however, no such instances were reported across centers. This information was obtained from standardized intraoperative surgical logs and operative reports, which were retrospectively reviewed for documentation of any deviation from the off-clamp protocol, including attempted or discussed clamping maneuvers. Additionally, a shared intraoperative protocol was in place across all centers to standardize surgical decision-making and hemostatic thresholds before clamping would be considered.

### 2.6. Perioperative Management

Perioperative care followed enhanced recovery protocols, including preoperative hydration, early mobilization, and pain control using multimodal analgesia. Patients were monitored postoperatively with serial vital signs, hemoglobin levels, serum creatinine, and drain output measurements. The drain was removed once output was minimal and non-hemorrhagic.

### 2.7. Data Collection

Demographic data collected included age, sex, body mass index (BMI), comorbidities (hypertension, diabetes, and chronic kidney disease), and American Society of Anesthesiologists (ASA) physical status classification.

Tumor characteristics recorded were size, location, side, and RENAL nephrometry score. Operative parameters included the following:Total operative time (from incision to skin closure),Console time (robot docking to undocking),Estimated blood loss (EBL),Need for intraoperative transfusion,Intraoperative complications,Conversion to hilar clamping or open surgery.

Postoperative outcomes comprised the following:Length of hospital stay,Complications graded by the Clavien–Dindo classification system,Readmission within 30 days,Pathological data including tumor histology, Fuhrman grade, surgical margin status,Imaging follow-up to assess recurrence.

### 2.8. Functional Outcome Assessment

Renal function was evaluated by measuring serum creatinine and eGFR preoperatively and postoperatively at 1 month, 3 months, and 12 months. Acute kidney injury (AKI) was defined according to the Kidney Disease Improving Global Outcomes (KDIGO) criteria, with staging based on changes in serum creatinine from baseline.

Renal function preservation was calculated as the percentage change in eGFR relative to baseline.

### 2.9. Oncologic Follow-Up

Patients underwent routine imaging (contrast-enhanced CT or MRI) at 3 months, 12 months, and annually thereafter to detect local recurrence or distant metastases. Positive surgical margin (PSM) was defined as tumor cells at the inked resection margin on final pathology.

### 2.10. Statistical Analysis

Data were analyzed using SPSS version 26.0 (IBM Corp., Armonk, NY, USA). Continuous variables were tested for normality using the Kolmogorov–Smirnov test. Normally distributed variables were reported as means ± standard deviations (SD), and non-normally distributed variables as medians with interquartile ranges (IQR).

Comparisons between groups were performed using Student’s *t*-test or Mann–Whitney U test for continuous variables, and Chi-square or Fisher’s exact test for categorical variables. Statistical significance was set at *p* < 0.05.

We performed a matching of the two groups based on both baseline (age, male:female distribution, BMI, ASA, AACCI, smoking status, and pre-operative eGFR) and pathological characteristics (focality, number of lesions, median tumor size, median renal score, and median Padua score) of the patients; the 2:1 ratio was used in order to highlight a marked difference, rather than the 1:1 ratio which is useful to highlight equality or balance.

## 3. Results

### 3.1. Patients’ Characteristics

Sample size: 563 patients in the off-clamp group vs. 244 in the on-clamp group (Table 1, Figure 1).Demographics and comorbidities: no statistically significant differences were noted between groups in the following:
Age at diagnosis (58 vs. 62 years)Gender distribution (approx. 70:30 male:female in both)ASA score (median 2 in both)Age-adjusted Charlson Comorbidity Index (AACCI; median 3 vs. 4)Smoking status (comparable proportions)Baseline renal function (eGFR median 87.4 vs. 85.9 mL/min/1.73 m^2^).

### 3.2. Pre-Operative Tumor Characteristics

Tumor features were statistically similar across both groups:
Focality: Predominantly monofocal in both (82% vs. 86%),Number of lesions: Median of 1 in both groups,Size of the largest lesion: 3.7 cm (off-clamp) vs. 4.2 cm (on-clamp),Complexity scores:
RENAL: Median 6 vs. 7,Padua: Median 6 vs. 7.

RENAL nephrometry scores were available in 95% of cases, with 52% low complexity (score 4–6), 38% moderate (score 7–9), and 10% high complexity (score ≥10).

### 3.3. Surgical Characteristics

Surgical approach: Robotic-assisted surgery predominated in both groups (97% vs. 86%), though the on-clamp group had slightly more laparoscopic/open cases.Length of surgery: Longer in the on-clamp group (118 vs. 140 min, *p* = 0.03).Estimated blood loss: Greater in the on-clamp group (150 vs. 200 mL, *p* = 0.03).Ischemia time: Only applicable to on-clamp (median 21 min).Complication rates:
Intraoperative: 0% (off-clamp) vs. 3.3% (on-clamp), *p* = 0.00.Perioperative: 9.2% (off-clamp) vs. 13.9% (on-clamp), *p* = 0.04.

In the off-clamp group, postoperative complications occurred in 52 patients (9.2%), the majority of which were Clavien–Dindo grade I–II. These included low-grade fever, transient hematuria, and postoperative ileus, all of which were managed conservatively without the need for invasive intervention.

Grade III complications were observed in nine patients (1.6%), primarily consisting of urinary leaks or perirenal collections requiring image-guided percutaneous drainage, and persistent urinary obstruction necessitating ureteral stent (double-J) placement. All grade III complications were managed minimally invasively.

Importantly, no Clavien–Dindo grade IV (life-threatening) or grade V (death) events occurred in this group. The median length of hospital stay was 3 days (range 2–7).

In comparison, the on-clamp group experienced a higher rate of postoperative complications, with a total of 34 patients (13.9%) affected. Among these, grade III complications occurred in 3.3% of cases, including a higher frequency of urinary leaks, bleeding requiring angioembolization, and obstructive events managed with drainage or endoscopic stenting. One Clavien–Dindo grade IV complication (0.4%) was recorded in the on-clamp group—a case of sepsis requiring intensive care admission. No deaths (grade V) were reported in either group.

The median length of hospital stay for off-clamp cases was 3 days (range: 2–7 days), reflecting a rapid recovery in most patients. The median length of stay in the on-clamp group was 4 days (range: 3–9 days), which was significantly longer compared to the off-clamp cohort (*p* < 0.05), likely reflecting higher complication burden.

### 3.4. Functional Outcomes

Baseline median eGFR was 87.4 mL/min/1.73 m^2^. At 1 month postoperatively, the median eGFR decline was 4.1%, with recovery plateauing by 3 months. No cases of stage 2 or higher AKI were observed in group 1. Comparatively, eGFR decline in controls undergoing on-clamp RAPN at the same institutions during the same period was significantly greater (mean decline 9.3%, *p* < 0.01) (Figure 2).

### 3.5. Oncologic Outcomes

pTNM stage: Comparable distribution (majority pT1a in both),Histology: Final pathology revealed clear cell RCC in 72%, papillary RCC in 18%, chromophobe RCC in 6%, and benign lesions (e.g., oncocytoma, angiomyolipoma) in 4%. Histological variants/aspects occurred in ~9% of both groups.Local recurrence (LR):
Rate: Low in both groups (0.36% vs. 1.2%, ns),Time to LR: Median 15 vs. 18 months.Positive surgical margins (PSM): Slightly higher in on-clamp group (3.7% vs. 2.3%, not significant).

### 3.6. Survival Outcomes

Overall survival (OS): 92.1% (off-clamp) vs. 87.8% (on-clamp),Cancer-specific survival (CSS): 95.4% vs. 93.3%,Both survival metrics were high and statistically comparable (*p* > 0.05).

## 4. Discussion

This large multicenter analysis reinforces the feasibility, safety, and clinical effectiveness of off-clamp RAPN in the management of localized renal tumors. By avoiding hilar clamping, this approach mitigates ischemia-induced renal injury—an important consideration particularly in patients with preexisting chronic kidney disease (CKD) or solitary kidneys—while maintaining favorable perioperative and oncologic outcomes.

Our findings show a modest postoperative decline in estimated glomerular filtration rate (eGFR), with a median loss of only 4.1% at 1 month and stabilization by 3 months. No patients experienced stage 3 or higher acute kidney injury (AKI) according to KDIGO criteria. These results underscore the renal functional benefit of eliminating warm ischemia, which is consistent with prior studies. Khalifeh et al. [6] and Bertolo et al. [12] previously reported superior renal function preservation with off-clamp RAPN, particularly in patients with baseline renal impairment. Moreover, the comparison with historical on-clamp RAPN controls from our own institutions revealed a significantly higher decline in eGFR (9.3%) versus the clampless cohort, further supporting the physiological advantages of ischemia avoidance.

The biological rationale behind the improved outcomes of clampless RAPN lies in avoiding the cascade of ischemia–reperfusion injury that begins once the renal artery is occluded. Warm ischemia, even when kept under the traditionally accepted 20–25-min threshold, has been shown to trigger oxidative stress, inflammation, and subsequent fibrosis in renal tissue, which can lead to irreversible nephron loss [14,15,16]. Clampless surgery circumvents this insult altogether, preserving microvascular integrity and nephron function. These benefits are especially pronounced in high-risk populations, including elderly patients, those with solitary kidneys, diabetes, hypertension, or other comorbidities predisposing to CKD [17,18,19].

The mechanistic context of off-clamp partial nephrectomy is expressed in two main areas: the renal protection and the recurrence biology. Regarding the renal protection, the off-clamp partial nephrectomy avoids temporary hilar clamping and subsequent warm ischemia, thereby preserving uninterrupted renal perfusion throughout the procedure. Mechanistically, this offers several nephroprotective advantages: avoidance of ischemia–reperfusion injury (clamping the renal hilum halts blood flow, causing cellular hypoxia and oxidative stress upon reperfusion; off-clamp PN eliminates this cycle, reducing tubular necrosis and inflammation); better preservation of nephron mass function: continuous perfusion allows better visualization of the natural tissue planes and may facilitate more precise excision, minimizing collateral parenchymal damage and safeguarding functional nephrons; improved early and long-term renal function: Avoiding ischemia particularly benefits patients with baseline renal impairment or solitary kidneys, as demonstrated by more favorable postoperative eGFR trends in clinical studies. Regarding the recurrence biology, while theoretical concerns have been raised regarding bleeding obscuring margins or tumor seeding in off-clamp procedures, current evidence—including from this dataset—does not support increased oncologic risk. Mechanistically: tumor handling and hemostasis (skilled surgical technique allows for controlled tumor excision even under perfused conditions, aided by improved imaging and robotic instrumentation); limited hypoxia-induced signaling (warm ischemia in on-clamp PN can activate hypoxia-inducible pathways—e.g., HIF-1α, which may theoretically promote angiogenesis and tumor aggressiveness; off-clamp PN circumvents this, potentially mitigating hypoxia-driven tumor progression); and low local recurrence rates: the observed equivalence in recurrence rates between off- and on-clamp groups (0.36% vs. 1.2%) reinforces the oncological safety of off-clamp surgery, even with more technically demanding conditions. In summary, off-clamp partial nephrectomy leverages continuous renal perfusion to minimize ischemic damage without compromising oncologic outcomes, which is supported by mechanistic and clinical evidence suggesting favorable preservation of renal function and no increase in recurrence risk.

The ischemia–reperfusion injury in the kidney activates a cascade of molecular events that significantly contribute to acute kidney injury (AKI) and promote the transition to chronic kidney disease. A key player in this process is the hypoxia-inducible factor-1 alpha (HIF-1α), which becomes stabilized under hypoxic conditions due to the inhibition of prolyl hydroxylases. Stabilized HIF-1α translocates to the nucleus and induces the transcription of genes involved in angiogenesis (e.g., VEGF), glycolysis, and inflammatory responses, potentially fostering maladaptive repair and fibrosis if persistently activated [20,21]. During reperfusion, reactive oxygen species (ROS) are rapidly generated by mitochondrial complexes, NADPH oxidases, and xanthine oxidase, overwhelming antioxidant defenses and causing oxidative damage to lipids, proteins, and DNA [22]. This oxidative burst exacerbates tubular cell apoptosis and necrosis, further compromising renal integrity. Concurrently, mitochondrial dysfunction, including disruption of membrane potential and impaired ATP synthesis, drives cellular energy failure and releases mitochondrial damage-associated molecular patterns (DAMPs), which trigger innate immune responses via toll-like receptors and the NLRP3 inflammasome [23]. These interconnected pathways—HIF-mediated signaling, ROS-induced injury, and mitochondrial damage—not only contribute to acute tubular injury, but also promote fibrogenesis through sustained activation of profibrotic cytokines such as TGF-β, leading to interstitial fibrosis and long-term loss of renal function [24]. Collectively, these mechanisms explain how transient ischemic insults during procedures, such as clamped partial nephrectomy, may accelerate the progression to CKD.

While our study showed favorable functional and oncologic outcomes in the off-clamp cohort, we acknowledge that the inclusion of validated biomarkers of renal injury or functional imaging metrics would have strengthened the mechanistic interpretation of nephron preservation.

In retrospect, surrogate serum or urinary biomarkers such as neutrophil gelatinase-associated lipocalin (NGAL) and kidney injury molecule-1 (KIM-1)—both established markers of early tubular injury—could have provided more granular insights into subclinical renal damage post-partial nephrectomy. Studies have shown that elevated NGAL and KIM-1 levels correlate with ischemia–reperfusion injury and predict the development of acute kidney injury and long-term renal dysfunction [25,26,27].

Furthermore, Doppler ultrasound-based renal resistive index (RRI) and advanced imaging tools, such as contrast-enhanced ultrasound, arterial spin labeling (ASL) MRI, or 99mTc-DMSA scintigraphy, have been explored in recent studies as non-invasive means of assessing renal perfusion and parenchymal preservation after nephron-sparing surgery [28,29]. Although such modalities were not incorporated into our current protocol, we agree that their application in future prospective studies could objectively capture dynamic changes in cortical perfusion and segmental nephron function, particularly in the early postoperative period.

We now acknowledged this limitation explicitly in the revised discussion and suggested the integration of such biomarkers and imaging proxies in future research efforts aiming to mechanistically validate the nephron-sparing advantage of off-clamp techniques.

Importantly, the improved functional outcomes observed with clampless RAPN were not offset by increased perioperative risks. Intraoperative complications occurred in only 1.2% of patients, transfusion rates were low (0.7%), and no case required conversion to hilar clamping or open surgery. Postoperative complications occurred in 9.2% of patients, mostly low-grade (Clavien–Dindo I–II), and were manageable with conservative or minimally invasive interventions. The overall safety profile is in line with, or better than, prior published series using on-clamp RAPN or laparoscopic approaches [12,13,30].

One key determinant of safety in clampless RAPN is effective hemostasis during tumor excision. This was achieved in our cohort through a combination of cold and energy-based dissection, selective use of sutures, and adjunctive hemostatic agents such as oxidized cellulose, gelatin matrix, and fibrin sealants. Previous studies have shown that when performed by experienced surgeons, clampless tumor resection can be achieved with acceptable bleeding risk and without compromising oncologic principles [31,32,33]. The ability to anticipate and control intraoperative bleeding is critical, particularly when resecting endophytic or hilar tumors where vascular anatomy is more complex.

Oncologic outcomes in our cohort were also favorable. The overall positive surgical margin (PSM) rate was 2.3%, which compares well with historical series of on-clamp RAPN, where PSM rates typically range from 1.6% to 4.3% [10,11,23]. Notably, the low recurrence rate (0.36%) at a median follow-up of 14 months reinforces the oncologic adequacy of tumor resection even in the absence of vascular clamping. This supports findings by Desai et al. [34] and Gill et al. [35], who demonstrated that enucleation and enucleoresection techniques during clampless RAPN can yield negative margins and excellent local control. While longer-term follow-up is necessary to fully validate these oncologic outcomes, our findings align with existing literature suggesting that warm ischemia avoidance does not compromise cancer control when tumors are well-visualized and carefully excised.

While it is intuitive to attribute longer operative time (OT) and greater estimated blood loss (EBL) in the on-clamp group to higher tumor complexity, our data do not support this explanation. In our cohort, the distribution of tumor complexity—assessed by both the RENAL nephrometry score and the PADUA score—was statistically comparable between the off-clamp and on-clamp groups (*p* > 0.05 for both scores). Specifically, the median RENAL score was 6 (IQR 5–8) in the off-clamp group versus 7 (IQR 6–8) in the on-clamp group, and the median PADUA score was 6 (IQR 5–8) versus 7 (IQR 6–9), respectively. Although these values suggest a numerically slightly higher complexity in the on-clamp cohort, the differences were not statistically significant, indicating that the two groups were well-matched in terms of tumor anatomical complexity. Therefore, the observed differences in OT and EBL likely reflect the procedural demands of hilar dissection, clamping, and subsequent reperfusion management in the on-clamp technique rather than inherent differences in tumor anatomy. This reinforces the validity of the comparison between the two surgical approaches in our analysis.

Regarding that 10% of tumors were high-complexity (RENAL > 10), the outcomes were comparable; these results were achieved in a cohort that included not only low- and moderate-complexity tumors, but also a significant proportion of high-complexity cases as defined by elevated RENAL and PADUA scores. This underscored the feasibility of off-clamp RAPN even in anatomically challenging scenarios, where concerns regarding intraoperative bleeding, positive surgical margins, or compromised visualization traditionally favor vascular clamping. The ability to safely perform clampless resection in such cases reflected advancements in surgical technique, refined robotic instrumentation, and growing surgeon experience. These findings suggested that off-clamp RAPN may be a viable nephron-sparing option across a broader spectrum of tumor complexities than previously assumed, offering functional preservation benefits without compromising surgical or oncologic integrity. Nonetheless, careful case selection and surgeon proficiency remain critical to maintaining safety and achieving optimal outcomes in more complex settings.

While our study stratified tumors using the RENAL score [36], a further clarification is warranted regarding its mechanistic association with perioperative risks, such as hemorrhage, renal functional decline, and incomplete tumor resection. Among the five components (radius—tumor size), exophytic/endophytic nature, nearness to the collecting system or sinus, anterior/posterior location, and location relative to polar lines) with which the RENAL nephrometry score quantifies tumor complexity, the nearness to sinus structures and endophytic location are particularly relevant mechanistically, as tumors in close proximity to the renal hilum or collecting system are more likely to disrupt or necessitate manipulation of segmental arteries and perisinusoidal microvasculature, increasing the risk of hemorrhage and ischemic injury during resection [37,38]. Higher RENAL scores also correlate with increased intraoperative blood loss, longer warm ischemia times, and a greater need for clamping, all of which are established contributors to postoperative renal functional deterioration [39,40]. From a microvascular standpoint, complex tumors often have distorted or intratumoral vasculature that impairs perfusion heterogeneity and elevates vascular resistance, rendering the surrounding parenchyma more susceptible to hypoxia and poor healing after surgery [41]. In addition, tumors with higher RENAL scores may pose a greater challenge in achieving negative surgical margins, particularly those that are endophytic or abutting the collecting system or sinus, due to limited visual and spatial access and higher likelihood of subclinical satellite nodules [42]. This has been associated with increased rates of positive surgical margins (PSMs) in some series. In light of this, a higher RENAL score inherently reflects more complex anatomical and microvascular tumor environments that may contribute to the risks of hemorrhage, functional loss, and incomplete resection, especially in off-clamp procedures where hemostasis is managed without vascular control.

The implications of our findings are significant. As surgical decision-making increasingly prioritizes functional preservation without compromising oncologic efficacy, clampless RAPN offers a strategy well-suited to the demands of modern nephron-sparing surgery. For selected patients, especially those with predisposing risk factors for renal decline or limited renal reserve, this approach may minimize the likelihood of postoperative CKD, reduce the need for renal replacement therapy in the long term, and improve overall quality of life [43,44,45].

A notable strength of this study lies in its multicenter design, encompassing three high-volume tertiary centers with consistent surgical protocols. The inclusion of 563 patients over a 6-year period allows for robust subgroup analyses and enhances generalizability to broader clinical practice. Furthermore, strict adherence to a clampless protocol provides a focused evaluation of this approach’s outcomes. Our findings thus offer one of the most comprehensive real-world assessments of clampless RAPN to date.

However, the study is not without limitations. The retrospective design inherently introduces selection bias and precludes randomization. Although data were collected prospectively, inherent differences in institutional protocols and follow-up intensity may have influenced outcome reporting.

Additionally, while early functional and oncologic results are promising, the median follow-up duration of 18 months remains insufficient to draw definitive conclusions regarding long-term outcomes. Renal function trajectories, particularly the potential for progressive decline due to subclinical ischemic injury, compensatory hyperfiltration, or parenchymal remodeling, often evolve over several years. Similarly, the risk of tumor recurrence—especially in patients with high-complexity lesions or adverse histological features—may not fully manifest within the first 1–2 years postoperatively. As such, the current follow-up period may underestimate the incidence of delayed renal impairment or late oncologic events. Longitudinal studies with extended surveillance are therefore essential to accurately assess the durability of renal function preservation, the risk of chronic kidney disease progression, and the true recurrence-free survival associated with clampless partial nephrectomy.

Furthermore, although the median 4.1% decrease in eGFR appeared minimal, it unfortunately did not include the use of the split renal scan methods to verify that there was overall preserved renal function (not just reverse compensation); this is an important limitation that could be addressed in the future by establishing more stringent criteria for assessing separate renal function.

Moreover, we did not perform subgroup analyses of functional outcomes in stratified populations according to baseline eGRF; this could negatively impact the interpretation of the results, so a subgroup analysis could certainly achieve better clinical applicability.

We attempted to limit multicenter heterogeneity through rigorous data collection; this could certainly represent a selection bias and could certainly be a limitation.

Future studies should aim to incorporate standardized follow-up imaging, quality-of-life metrics, and longer-term endpoints beyond five years.

Furthermore, patient selection for clampless RAPN remains a nuanced decision, often guided by tumor characteristics such as size, location, proximity to vascular structures, and RENAL nephrometry score. In our series, more than half of the tumors were low-complexity lesions; while we included moderate and high-complexity cases, further research is needed to delineate the safety and efficacy of clampless techniques in these subgroups. Emerging adjuncts such as near-infrared fluorescence imaging with indocyanine green, 3D imaging reconstruction, and augmented reality platforms may enhance intraoperative visualization and further extend the indications for clampless approaches [46,47].

In summary, clampless RAPN offers a compelling strategy to maximize renal function preservation without compromising perioperative safety or oncologic control. Its broader adoption will depend on further prospective data, longer-term follow-up, and wider dissemination of technical expertise. As surgical technology evolves and our understanding of renal preservation deepens, the clampless paradigm may represent a pivotal advancement in the management of localized renal tumors.

## 5. Conclusions

Clampless robot-assisted partial nephrectomy appears to be a technically feasible and oncologically safe nephron-sparing approach for select patients with localized renal tumors. In our multicenter experience, the technique was associated with favorable short- to mid-term renal functional preservation and a low rate of complications, without compromising early oncologic outcomes. However, given the limited duration of follow-up and the absence of long-term renal and cancer-specific endpoints, these findings should be interpreted with caution. Broader implementation of clampless techniques should be guided by careful patient selection, particularly in individuals at elevated risk for renal insufficiency. Further prospective studies with extended follow-up are warranted to confirm these initial results and to better define the durability of functional and oncologic outcomes over time.

## Figures and Tables

**Figure 1 cancers-17-02645-f001:**
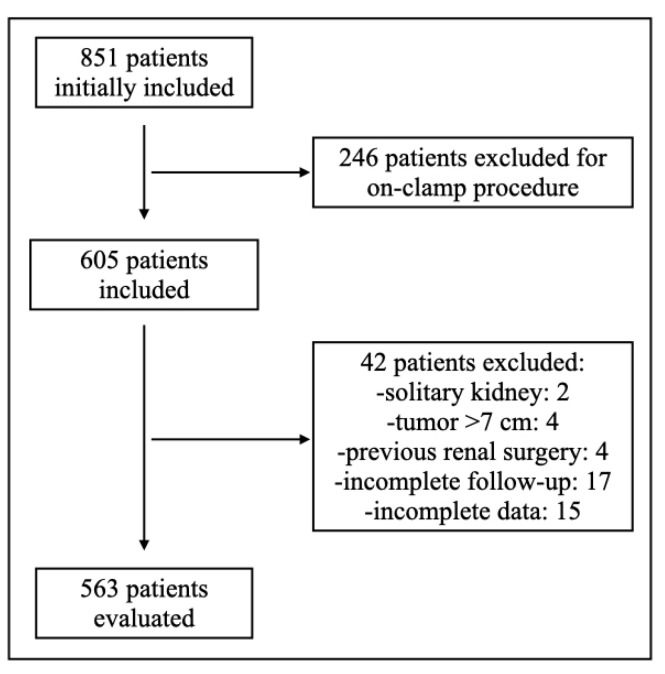
Flow chart.

**Figure 2 cancers-17-02645-f002:**
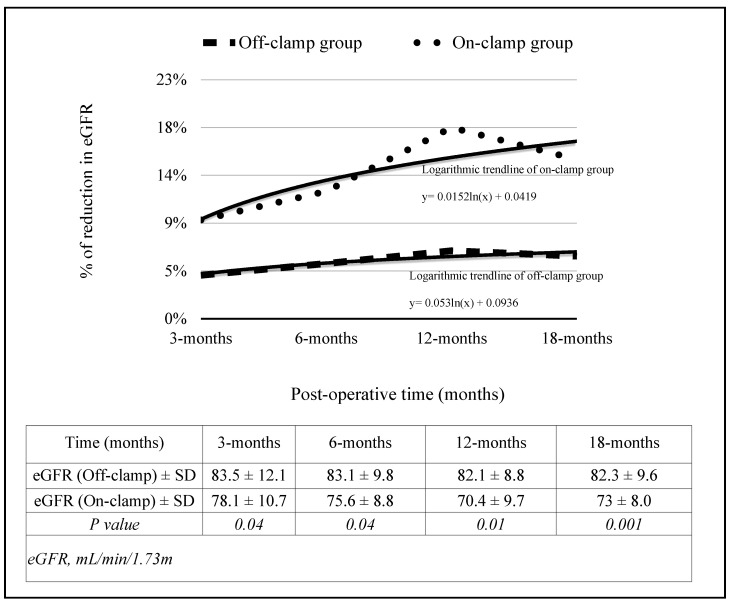
Trend of GFR-loss.

**Table 1 cancers-17-02645-t001:** Summary table.

	Off-Clamp	On-Clamp	*p*-Value
Patients’ characteristics			>0.05
Number of patients	563	244	
Age at diagnosis, years	58 (35–80)	62 (55–73)	
Gender, m:f, %	70.9:29.1	73.3:26.7	
ASA score, median (range)	2 (2–3)	2 (2–3)	
AACCI, median (range)	3 (3–5)	4 (4–5)	
Smoking status, no smoker: active/former smoker, %	64.5:35.5	69.7:30.3	
Pre-operative eGFR, mL/min/1.73 m^2^, median (IQR)	87.4 (57.9–108.3)	85.9 (38–98)	
Pre-operative tumour characteristics			>0.05
Focality, monofocal:multifocal, %	81.9:18.1	85.7:14.3	
N° of lesions, median (range)	1 (1–2)	1 (1–3)	
Median size of the largest lesion, cm (IQR)	3.7 (1.0–5.8)	4.2 (2.3–5.3)	
Median RENAL score, median value (IQR)	6 (5–8)	7 (6–8)	
Median Padua score, median (IQR)	6 (5–8)	7 (6–9)	
Surgical characteristics			
Approach, robotic:laparoscopic:open	96.8:3.2:0	86.1:9:4.9	>0.05
Median length of surgery, minutes (IQR)	118 (100–140)	140 (120–182)	0.03
Median blood loss, mL (IQR)	150 (50–400)	200 (180–300)	0.03
Median ischemia time, minutes (IQR)	-	21 (15.5–24)	-
Intraoperative complications, %	0	3.3	0.00
Perioperative complications, %	9.2	13.9	0.04
Pathological characteristics			>0.05
pTNM			
pT1a	78.9%	81.5%	
pT1b	21.1%	18.5%	
Histology			
Clear cell RCC	85.7%	83.6%	
Papillary RCC	12.3%	11.9%	
Chromophobe RCC	2.1%	4.5%	
Variants/aspects, %	8.9%	9.8%	
Rate of LR, %	0.36	1.2	>0.05
Median time to LR, months (IQR)	15 (13–19)	18 (12–39)	>0.05
Rate of PSM, %	2.3	3.7	>0.05
OS, %	92.1	87.8	>0.05
CSS, %	95.4	93.3	>0.05

ASA: American Society of Anesthesiologists, AACCI: adjusted age-adjusted Charlson comorbidity index, eGFR: estimated glomerular filtration rate, RENAL: nephrometry score, RCC: renal cell carcinoma, LR: local recurrence, PSM: positive surgical margin, OS: overall survival, and CSS: cancer specific survival.

## Data Availability

The data presented in this study are available on request from the corresponding author.

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
