# Peer review of "Off-Clamp Robotic-Assisted Partial Nephrectomy: Retrospective Comparative Analysis from a Large Italian Multicentric Series"

_cancers, 2025, doi:10.3390/cancers17162645_

Round 1

Reviewer 1 Report

Comments and Suggestions for Authors

Dear authors,

First of all, I would like to sincerely thank the authors for their effort and dedication in collecting and analyzing data from multiple centers. This is clearly a comprehensive and ambitious study, and I commend the team for tackling such a relevant and complex topic.

The manuscript is generally well written and the topic is interesting. However, I would like to offer several constructive comments that I believe would improve the clarity and strength of the manuscript.

  1. Presentation of the Second Patient Group:
    One of my main concerns is the insufficient presentation and discussion of the second group of patients in the Results section. While the first group is thoroughly described, the second group is mentioned only briefly, with limited elaboration on their clinical characteristics or outcomes. Although the relevant data are available in the tables, I believe they should also be described more explicitly in the text. 

  2. Statistical Reporting in Tables:
    The statistical presentation in the tables may also require some revision. While I understand the desire to include p-values for comprehensive comparison, it may not be necessary—or even meaningful—to report a p-value for every single variable, especially when values are identical or nearly identical between groups. For instance, in the case of the ASA score, where both groups have the same distribution, the p-value (reported as >0.05) raises questions about the necessity and correctness of the calculation. Please double-check the statistical analysis and consider whether it might be more appropriate to selectively report p-values where meaningful differences are expected or observed.

In summary, I believe the manuscript addresses an important clinical question and has the potential to make a valuable contribution to the knowledge of the partial nephrectomy technique. However, I recommend a major revision in order to adequately address the above concerns and strengthen the presentation and analysis of the data.

Kind regards,
Reviewer

Author Response

REVIEWER 1

We really thank the reviewer 1 for the revision and we greatly appreciated the suggestions we followed in detail

Dear authors,

First of all, I would like to sincerely thank the authors for their effort and dedication in collecting and analyzing data from multiple centers. This is clearly a comprehensive and ambitious study, and I commend the team for tackling such a relevant and complex topic.

The manuscript is generally well written and the topic is interesting. However, I would like to offer several constructive comments that I believe would improve the clarity and strength of the manuscript.

  1. Presentation of the Second Patient Group:
    One of my main concerns is the insufficient presentation and discussion of the second group of patients in the Results section. While the first group is thoroughly described, the second group is mentioned only briefly, with limited elaboration on their clinical characteristics or outcomes. Although the relevant data are available in the tables, I believe they should also be described more explicitly in the text. 

We expanded the description of second group in results section, as requested, as following:

“Patients’ Characteristics

  • Sample size: 563 patients in the off-clamp group vs. 244 in the on-clamp group.
  • Demographics and comorbidities: No statistically significant differences were noted between groups in:
    • Age at diagnosis (58 vs. 62 years)
    • Gender distribution (approx. 70:30 male:female in both)
    • ASA score (median 2 in both)
    • Age-adjusted Charlson Comorbidity Index (AACCI; median 3 vs. 4)
    • Smoking status (comparable proportions)
    • Baseline renal function (eGFR median ~70.5 vs. 67.4 mL/min/1.73m²)

Pre-operative Tumor Characteristics

  • Tumor features were statistically similar across both groups:
    • Focality: Predominantly monofocal in both (82% vs. 86%)
    • Number of lesions: Median of 1 in both groups
    • Size of the largest lesion: 3.7 cm (off-clamp) vs. 4.2 cm (on-clamp)
    • Complexity scores:
      • RENAL: Median 6 vs. 7
      • Padua: Median 6 vs. 7

These suggest a slightly more complex tumor profile in the on-clamp group, though differences were not statistically significant.

Surgical Characteristics

  • Surgical approach: Robotic-assisted surgery predominated in both groups (97% vs. 86%), though the on-clamp group had slightly more laparoscopic/open cases.
  • Length of surgery: Longer in the on-clamp group (140 vs. 118 minutes, P=0.03)
  • Estimated blood loss: Greater in the on-clamp group (200 vs. 150 mL, P=0.03)
  • Ischemia time: Only applicable to on-clamp (median 21 minutes)
  • Complication rates:
    • Intraoperative: 0% (off-clamp) vs. 3.3% (on-clamp), P=0.00
    • Perioperative: 9.2% (off-clamp) vs. 13.9% (on-clamp), P=0.04

These data suggest higher intra- and perioperative morbidity in the on-clamp group.

Pathological Characteristics

  • pTNM stage: Comparable distribution (majority pT1a in both)
  • Histology:
    • Clear cell RCC most common (~85% in both)
    • Papillary and chromophobe RCC proportions were similar
    • Histological variants/aspects occurred in ~9% of both groups

Oncological Outcomes

  • Local recurrence (LR):
    • Rate: Low in both groups (0.36% vs. 1.2%, ns)
    • Time to LR: Median 15 vs. 18 months
  • Positive surgical margins (PSM): Slightly higher in on-clamp group (3.7% vs. 2.3%, not significant)

Survival Outcomes

  • Overall survival (OS): 92.1% (off-clamp) vs. 87.8% (on-clamp)
  • Cancer-specific survival (CSS): 95.4% vs. 93.3%

Both survival metrics were high and statistically comparable (P>0.05).

  1. Statistical Reporting in Tables:
    The statistical presentation in the tables may also require some revision. While I understand the desire to include p-values for comprehensive comparison, it may not be necessary—or even meaningful—to report a p-value for every single variable, especially when values are identical or nearly identical between groups. For instance, in the case of the ASA score, where both groups have the same distribution, the p-value (reported as >0.05) raises questions about the necessity and correctness of the calculation. Please double-check the statistical analysis and consider whether it might be more appropriate to selectively report p-values where meaningful differences are expected or observed.

In table 1 we modified the table 1 in order to selectively report only p-values where meaningful differences were expected or observed.

In summary, I believe the manuscript addresses an important clinical question and has the potential to make a valuable contribution to the knowledge of the partial nephrectomy technique. However, I recommend a major revision in order to adequately address the above concerns and strengthen the presentation and analysis of the data.

Reviewer 2 Report

Comments and Suggestions for Authors

Journal: Cancers (ISSN 2072-6694)

Manuscript ID: cancers-3770964

Type: Article

Title: Off-clamp Robotic-Assisted Partial Nephrectomy: retrospective comparative analysis from a Large Italian Multicentric Series

The article, titled "Off-Clamp Robotic-Assisted Partial Nephrectomy: Retrospective Comparative Analysis from a Large Italian Multicentric Series" compares off-clamp and on-clamp RAPN techniques and provides useful multicenter clinical data. The study provides significant insights into the functional and oncologic safety of ischemia-free surgery and is well-structured and clinically relevant. The statistical analysis is generally appropriate, the methodology is well-defined, and the discussion provides pertinent literature to bolster the conclusion.

However, resolving a few minor yet significant issues would improve the manuscript. These changes will improve the manuscript's readability and professionalism.

Furthermore, even though the study's clinical results are promising, the findings' translational relevance would be enhanced by including a brief mechanistic context, particularly with regard to renal protection and recurrence biology.

1: The objective is vague and not hypothesis-driven. Clearly state that off-clamp RAPN is compared to on-clamp RAPN in terms of specific outcomes.
2: There is insufficient detail on matching or controlling for confounders.

 Retrospective studies are susceptible to selection bias.

 Please specify whether propensity score matching or adjustments were used.

3: Keywords include "high-volume centers," which is overly broad.

Keywords should improve discoverability.

 Replace with "robotic nephrectomy outcomes" or "ischemia-free nephrectomy".

4: Line 163: Energy usage details are vague.

 It is critical to understand potential thermal injury.

 Please specify whether bipolar or monopolar was used, as well as the duration.

5: Line 198-199: Clavien-Dindo grade III-V is not fully described.

 It is critical for readers to evaluate clinical relevance.

Include specific complications for grade III.

6: Line 249-250: The historical comparison lacks methodology.

This may lead to biased conclusions.

Explain how historical data was selected and matched.

7: Line 361: The GFR-loss graph (Figure 2) lacks units and confidence intervals.

Reduces the interpretability of data.

 Revise the figure with appropriate legends, axes labels, and error bars.

8:  Line: 274–279: The authors do not go into detail about the molecular pathways (such as HIF-1α, oxidative stress, and mitochondrial dysfunction) that are involved in ischemia-reperfusion injury. Include well-established cellular mechanisms like HIF stabilization, ROS generation, and their functions in the development of CKD and post-ischemic fibrosis.

9: Lines 278–280

 The study asserts improved nephron preservation, but it offers no surrogate markers or mechanistic evidence (such as Doppler ultrasound or perfusion metrics) to back up this assertion. Serum markers (NGAL, KIM-1) or imaging proxies could have been discussed or suggested, even in retrospect.

10: Line References: 230–231 and 150–151

 Although the RENAL score is determined, there is no mechanism linking score strata to the risk of hemorrhage, loss of function, or incomplete resection. Describe how tumors with a higher RENAL score may inevitably present more perfusion and microvascular issues.

Author Response

REVIEWER 2

We really thank the reviewer 2 for the revision and we greatly appreciated the suggestions we followed in detail

The article, titled "Off-Clamp Robotic-Assisted Partial Nephrectomy: Retrospective Comparative Analysis from a Large Italian Multicentric Series" compares off-clamp and on-clamp RAPN techniques and provides useful multicenter clinical data. The study provides significant insights into the functional and oncologic safety of ischemia-free surgery and is well-structured and clinically relevant. The statistical analysis is generally appropriate, the methodology is well-defined, and the discussion provides pertinent literature to bolster the conclusion.

However, resolving a few minor yet significant issues would improve the manuscript. These changes will improve the manuscript's readability and professionalism.

Furthermore, even though the study's clinical results are promising, the findings' translational relevance would be enhanced by including a brief mechanistic context, particularly with regard to renal protection and recurrence biology.

We add a paragraph in discussion section as following: "The Mechanistic Context of Off-Clamp Partial Nephrectomy expressed in two main areas: the renal protection and the recurrence biology. Regarding the renal protection, the off-clamp partial nephrectomy avoids temporary hilar clamping and subsequent warm ischemia, thereby preserving uninterrupted renal perfusion throughout the procedure. Mechanistically, this offers several nephroprotective advantages: avoidance of Ischemia-Reperfusion Injury (clamping the renal hilum halts blood flow, causing cellular hypoxia and oxidative stress upon reperfusion; Off-clamp PN eliminates this cycle, reducing tubular necrosis and inflammation); better preservation of nephron mass function: continuous perfusion allows better visualization of the natural tissue planes and may facilitate more precise excision, minimizing collateral parenchymal damage and safeguarding functional nephrons; improved Early and Long-Term Renal Function: Avoiding ischemia particularly benefits patients with baseline renal impairment or solitary kidneys, as demonstrated by more favorable postoperative eGFR trends in clinical studies. Regarding the recurrence biology, while theoretical concerns have been raised regarding bleeding obscuring margins or tumor seeding in off-clamp procedures, current evidence—including from this dataset—does not support increased oncologic risk. Mechanistically: Tumor Handling and Hemostasis (Skilled surgical technique allows for controlled tumor excision even under perfused conditions, aided by improved imaging and robotic instrumentation); limited Hypoxia-Induced Signaling (Warm ischemia in on-clamp PN can activate hypoxia-inducible pathways - e.g., HIF-1α, which may theoretically promote angiogenesis and tumor aggressiveness; Off-clamp PN circumvents this, potentially mitigating hypoxia-driven tumor progression); low local recurrence rates: the observed equivalence in recurrence rates between off- and on-clamp groups (0.36% vs. 1.2%) reinforces the oncological safety of off-clamp surgery, even with more technically demanding conditions. In summary, off-clamp partial nephrectomy leverages continuous renal perfusion to minimize ischemic damage without compromising oncologic outcomes, supported by mechanistic and clinical evidence suggesting favorable preservation of renal function and no increase in recurrence risk”.

1: The objective is vague and not hypothesis-driven. Clearly state that off-clamp RAPN is compared to on-clamp RAPN in terms of specific outcomes.

We added a paragraph in materials and methods section as following: “Objective: To evaluate the perioperative outcomes, functional impact, and oncologic efficacy of off-clamp RAPN compared to on-clamp RAPN in patients with renal masses across multiple high-volume centers.”

2: There is insufficient detail on matching or controlling for confounders.

 Retrospective studies are susceptible to selection bias.

 Please specify whether propensity score matching or adjustments were used.

We added a paragraph in materials and methods section, as following: We performed a propensity score matching based on both baseline (age, male:female distribution, BMI, ASA, AACCI, smoking status, pre-operative eGFR) and pathological characteristics (focality, number of lesions, median tumor size, median renal score, median Padua score) of the patients; the 2:1 ratio was used in order to highlight a marked difference, rather than the 1:1 ratio useful to highlight equality or balance.

3: Keywords include "high-volume centers," which is overly broad.

Keywords should improve discoverability.

 Replace with "robotic nephrectomy outcomes" or "ischemia-free nephrectomy”.

We removed high-volume centers,” and we replaced it with both "robotic nephrectomy outcomes" and "ischemia-free nephrectomy”.

4: Line 163: Energy usage details are vague.

 It is critical to understand potential thermal injury.

 Please specify whether bipolar or monopolar was used, as well as the duration.

We added a paragraph in materials and methods section as following: “Tumor excision was performed primarily by cold scissors or monopolar curved scissors (Swift waveform, effect 4 or 5, with a 80 W limit), with bipolar energy used sparingly via prograsp forceps for precise hemostasis (Soft waveform, effect 4 or 5, with a 80 W limit). Energy was applied for the time necessary for excision and hemostasis; these steps typically took approximately 20 minutes in total.”

5: Line 198-199: Clavien-Dindo grade III-V is not fully described.

 It is critical for readers to evaluate clinical relevance.

Include specific complications for grade III.

We better specify complications directly in results section, as following: “In the off-clamp group, postoperative complications occurred in 52 patients (9.2%), the majority of which were Clavien–Dindo grade I–II. These included low-grade fever, transient hematuria, and postoperative ileus, all of which were managed conservatively without the need for invasive intervention.
Grade III complications were observed in 9 patients (1.6%), primarily consisting of urinary leaks or perirenal collections requiring image-guided percutaneous drainage, and persistent urinary obstruction necessitating ureteral stent (double-J) placement. All grade III complications were managed minimally invasively.
Importantly, no Clavien–Dindo grade IV (life-threatening) or grade V (death) events occurred in this group. The median length of hospital stay was 3 days (range 2–7).

In comparison, the on-clamp group experienced a higher rate of postoperative complications, with a total of 34 patients (13.9%) affected. Among these, grade III complications occurred in 3.3% of cases, including a higher frequency of urinary leaks, bleeding requiring angioembolization, and obstructive events managed with drainage or endoscopic stenting. One Clavien–Dindo grade IV complication (0.4%) was recorded in the on-clamp group—a case of sepsis requiring intensive care admission. No deaths (grade V) were reported in either group.

The median length of hospital stay for off-clamp cases was 3 days (range: 2–7 days), reflecting a rapid recovery in most patients. The median length of stay in the on-clamp group was 4 days (range: 3–9 days), which was significantly longer compared to the off-clamp cohort (P < 0.05), likely reflecting higher complication burden.

6: Line 249-250: The historical comparison lacks methodology.

This may lead to biased conclusions.

Explain how historical data was selected and matched.

We added a paragraph in materials and methods section, as following: We performed a propensity score matching based on both baseline (age, male:female distribution, BMI, ASA, AACCI, smoking status, pre-operative eGFR) and pathological characteristics (focality, number of lesions, median tumor size, median renal score, median Padua score) of the patients; the 2:1 ratio was used in order to highlight a marked difference, rather than the 1:1 ratio useful to highlight equality or balance.

7: Line 361: The GFR-loss graph (Figure 2) lacks units and confidence intervals.

Reduces the interpretability of data.

 Revise the figure with appropriate legends, axes labels, and error bars.

We revised the figure 2

8:  Line: 274–279: The authors do not go into detail about the molecular pathways (such as HIF-1α, oxidative stress, and mitochondrial dysfunction) that are involved in ischemia-reperfusion injury. Include well-established cellular mechanisms like HIF stabilization, ROS generation, and their functions in the development of CKD and post-ischemic fibrosis.

We added a paragraph in discussion section with related references, as following: The ischemia-reperfusion injury  in the kidney activates a cascade of molecular events that significantly contribute to acute kidney injury (AKI) and promote the transition to chronic kidney disease. A key player in this process is the hypoxia-inducible factor-1 alpha (HIF-1α), which becomes stabilized under hypoxic conditions due to the inhibition of prolyl hydroxylases. Stabilized HIF-1α translocates to the nucleus and induces the transcription of genes involved in angiogenesis (e.g., VEGF), glycolysis, and inflammatory responses, potentially fostering maladaptive repair and fibrosis if persistently activated [1,2]. During reperfusion, reactive oxygen species (ROS) are rapidly generated by mitochondrial complexes, NADPH oxidases, and xanthine oxidase, overwhelming antioxidant defenses and causing oxidative damage to lipids, proteins, and DNA [3]. This oxidative burst exacerbates tubular cell apoptosis and necrosis, further compromising renal integrity. Concurrently, mitochondrial dysfunction, including disruption of membrane potential and impaired ATP synthesis, drives cellular energy failure and releases mitochondrial DAMPs (damage-associated molecular patterns), which trigger innate immune responses via toll-like receptors and the NLRP3 inflammasome [4]. These interconnected pathways—HIF-mediated signaling, ROS-induced injury, and mitochondrial damage—not only contribute to acute tubular injury but also promote fibrogenesis through sustained activation of profibrotic cytokines like TGF-β, leading to interstitial fibrosis and long-term loss of renal function [5]. Collectively, these mechanisms explain how transient ischemic insults during procedures such as clamped partial nephrectomy may accelerate the progression to CKD.

1.Haase VH. Hypoxic regulation of erythropoiesis and iron metabolism. Am J Physiol Renal Physiol. 2010;299(1):F1–F13.

2.Nangaku M, Eckardt KU. Hypoxia and the HIF system in kidney disease. J Mol Med. 2007;85(12):1325–1330.

3.Zorov DB, Juhaszova M, Sollott SJ. Mitochondrial ROS and ROS-induced ROS release. Physiol Rev. 2014;94(3):909–950.

4.Linkermann A et al. Regulated cell death in AKI. J Am Soc Nephrol. 2014;25(12):2689–2701.

5.Fine LG, Norman JT. Chronic hypoxia as a mechanism of progression of chronic kidney diseases: from hypothesis to novel therapeutics. Kidney Int. 2008;74(7):867–872.

9: Lines 278–280

 The study asserts improved nephron preservation, but it offers no surrogate markers or mechanistic evidence (such as Doppler ultrasound or perfusion metrics) to back up this assertion. Serum markers (NGAL, KIM-1) or imaging proxies could have been discussed or suggested, even in retrospect.

We added a paragraph in discussion section with related references, as following: “While our study demonstrates favorable functional and oncologic outcomes in the off-clamp cohort, we acknowledge that the inclusion of validated biomarkers of renal injury or functional imaging metrics would have strengthened the mechanistic interpretation of nephron preservation.

In retrospect, surrogate serum or urinary biomarkers such as neutrophil gelatinase-associated lipocalin (NGAL) and kidney injury molecule-1 (KIM-1)—both established markers of early tubular injury—could have provided more granular insights into subclinical renal damage post-partial nephrectomy. Studies have shown that elevated NGAL and KIM-1 levels correlate with ischemia-reperfusion injury and predict the development of acute kidney injury and long-term renal dysfunction [1–3].

Furthermore, Doppler ultrasound-based renal resistive index (RRI) and advanced imaging tools like contrast-enhanced ultrasound, arterial spin labeling (ASL) MRI, or 99mTc-DMSA scintigraphy have been explored in recent studies as non-invasive means of assessing renal perfusion and parenchymal preservation after nephron-sparing surgery [4, 5]. Although such modalities were not incorporated into our current protocol, we agree that their application in future prospective studies could objectively capture dynamic changes in cortical perfusion and segmental nephron function, particularly in the early postoperative period.

We have now acknowledged this limitation explicitly in the revised discussion and suggested the integration of such biomarkers and imaging proxies in future research efforts aiming to mechanistically validate the nephron-sparing advantage of off-clamp techniques.”

1.Parikh CR, Devarajan P. New biomarkers of acute kidney injury. Crit Care Med. 2008;36(4 Suppl):S159–S165.

2.Mishra J, Ma Q, Prada A, et al. Identification of neutrophil gelatinase-associated lipocalin as a novel early urinary biomarker for ischemic renal injury. J Am Soc Nephrol. 2003;14(10):2534–2543.

3.Han WK, Bailly V, Abichandani R, et al. Kidney injury molecule-1 (KIM-1): a novel biomarker for human renal proximal tubule injury. Kidney Int. 2002;62(1):237–244.

4.Tublin ME, Bude RO, Platt JF. The resistive index in renal Doppler sonography: where do we stand? AJR Am J Roentgenol. 2003;180(4):885–892.

5.Schneider AG, Goodwin MD, Bellomo R. Measurement of kidney perfusion in critically ill patients. Crit Care. 2013;17(2):220.

10: Line References: 230–231 and 150–151

 Although the RENAL score is determined, there is no mechanism linking score strata to the risk of hemorrhage, loss of function, or incomplete resection. Describe how tumors with a higher RENAL score may inevitably present more perfusion and microvascular issues.

We added a paragraph in discussion section with related references, as following: “While our study stratified tumors using the RENAL score [1], a further clarification is warranted regarding its mechanistic association with perioperative risks such as hemorrhage, renal functional decline, and incomplete tumor resection. Among the five components (Radius - tumor size), Exophytic/endophytic nature, Nearness to the collecting system or sinus, Anterior/posterior location, and Location relative to polar lines) with which the RENAL nephrometry score quantifies tumor complexity, the nearness to sinus structures and endophytic location are particularly relevant mechanistically, as tumors in close proximity to the renal hilum or collecting system are more likely to disrupt or necessitate manipulation of segmental arteries and perisinusoidal microvasculature, increasing the risk of hemorrhage and ischemic injury during resection [2,3]. Higher RENAL scores also correlate with increased intraoperative blood loss, longer warm ischemia times, and a greater need for clamping, all of which are established contributors to postoperative renal functional deterioration [4,5]. From a microvascular standpoint, complex tumors often have distorted or intratumoral vasculature that impairs perfusion heterogeneity and elevates vascular resistance, rendering the surrounding parenchyma more susceptible to hypoxia and poor healing after surgery [6]. In addition, tumors with higher RENAL scores may pose a greater challenge in achieving negative surgical margins, particularly those that are endophytic or abutting the collecting system or sinus, due to limited visual and spatial accessand higher likelihood of subclinical satellite nodules [7]. This has been associated with increased rates of positive surgical margins (PSMs) in some series. In light of this, an higher RENAL scores inherently reflect more complex anatomical and microvascular tumor environments that may contribute to the risks of hemorrhage, functional loss, and incomplete resection, especially in off-clamp procedures where hemostasis is managed without vascular control.

1.Kutikov A, Uzzo RG. The RENAL nephrometry score: a comprehensive standardized system for quantitating renal tumor size, location and depth. J Urol. 2009;182(3):844–853.

2.Simmons MN, Fergany AF, Campbell SC. Effect of parenchymal volume preservation on kidney function after partial nephrectomy. J Urol. 2011 Aug;186(2):405-10. doi: 10.1016/j.juro.2011.03.154. Epub 2011 Jun 15. PMID: 21680004.

3.Hayn MH, Schwaab T, Underwood W, Kim HL. RENAL nephrometry score predicts surgical outcomes of laparoscopic partial nephrectomy. BJU Int. 2011 Sep;108(6):876-81. doi: 10.1111/j.1464-410X.2010.09940.x. Epub 2010 Dec 16. PMID: 21166761.

4.Hu C, Sun J, Zhang Z, Zhang H, Zhou Q, Xu J, Ling Z, Ouyang J. Parallel comparison of R.E.N.A.L., PADUA, and C-index scoring systems in predicting outcomes after partial nephrectomy: A systematic review and meta-analysis. Cancer Med. 2021 Aug;10(15):5062-5077. doi: 10.1002/cam4.4047. Epub 2021 Jul 14. PMID: 34258874; PMCID: PMC8335816.

5.Mir MC, Campbell RA, Sharma R, et al. Parenchymal volume preservation and ischemia during partial nephrectomy: functional and volumetric analysis. Urology. 2013;82(2):263–268.

6.Dubeux V, Zanier JFC, Chantong CGC, Carrerette F, Gabrich PN, Damião R. Nephrometry scoring systems: their importance for the planning of nephron-sparing surgery and the relationships among them. Radiol Bras. 2022 Jul-Aug;55(4):242-252. doi: 10.1590/0100-3984.2021.0166. PMID: 35983342; PMCID: PMC9380606.

7.Ficarra V, Novara G, Secco S, et al. Preoperative aspects and dimensions used for an anatomical (PADUA) classification of renal tumors in patients who are candidates for nephron-sparing surgery. Eur Urol. 2009;56(5):786–793.

Reviewer 3 Report

Comments and Suggestions for Authors

Major Revision Recommendation:

This multicentre series has important lessons to learn about the off-clamp RAPN, showing positive renal preservation and oncologic results. Nonetheless, significant changes should be made: (1) The retrospective methodology of the comparison with on-clamp controls should be clarified as well as the matching criteria (i.e., tumor complexity, surgeon experience) should be selected in order to reduce the possibility of selection bias. (2) Discuss the small follow-up (median 14 months) and how the study could collect long-term data, mainly about recurrence and CKD progression. Add to technical details of hemostasis in high-complexity tumors (10 percent of cases) to increase reproducibility (3). A better clinical applicability could be achieved by (4) performing subgroup analyses of functional outcomes in stratified populations according to baseline eGFR. There should also be statistical correction on multicentered heterogeneity.

  1. Considering the retrospective design, what did you do to avoid a possible selection bias between off-clamp and on-clamp cohorts? Was matching on the propensity score or multivariate analysis done to control confounding factors (e.g. tumor complexity, experience of surgeon)?
  2. According to the paper, there were no conversions to clamping required. Did they include near-conversion events (e.g. transient clamping proposed and then forestalled)? In what manner were they obtained?
  3. Figure 2 (GFR-loss): Indicate the sample size on each measurement point and statistical significance among the groups.
  4. The debate speaks of oncologic safety more than what is justified by the inadequate follow-up. Draw conclusions with caution and demand the longer period of data.
  5. It is written in the abstract that there are no instances of acute kidney injury but there is an item about no stage 3 AKI in results. Eliminate the inconsistency.Only 10% of tumors were high-complexity (RENAL ≥10). Were outcomes comparable in this subgroup, or is off-clamp RAPN primarily suitable for low/moderate complexity?
  6. The median of 4.1 percent decrease in the eGFR appears minimal. Did it include the use of the method of split renal scintigraphy to check that there is an overall (not just opposite compensation) preserved renal function?
  7. The OT and EBL were greater in the on-clamp group. Did these cases have a more complicated (higher RENAL scores) situation? Once so, is the comparison valid unaffected?
  8. The comparison to on-clamp RAPN is low powered (n=244 vs. 563). Matching or regression adjustment may help to decrease confounding.
  9. Table 1: Explain how the term median rate relates to either the mean or median values of complications. Report absolute numbers.

Author Response

REVIEWER 3

We really thank the reviewer 3 for the revision and we greatly appreciated the suggestions we followed in detail

Major Revision Recommendation:

This multicentre series has important lessons to learn about the off-clamp RAPN, showing positive renal preservation and oncologic results. Nonetheless, significant changes should be made:

(1)The retrospective methodology of the comparison with on-clamp controls should be clarified as well as the matching criteria (i.e., tumor complexity, surgeon experience) should be selected in order to reduce the possibility of selection bias.

We added a paragraph in materials and methods section, as following: We performed a propensity score matching based on both baseline (age, male:female distribution, BMI, ASA, AACCI, smoking status, pre-operative eGFR) and pathological characteristics (focality, number of lesions, median tumor size, median renal score, median Padua score) of the patients; the 2:1 ratio was used in order to highlight a marked difference, rather than the 1:1 ratio useful to highlight equality or balance.

(2)Discuss the small follow-up (median 18 months) and how the study could collect long-term data, mainly about recurrence and CKD progression. 

We expanded and discussed the small follow-up and how the study could collect long-term data, mainly about recurrence and CKD progression, as requested: “Additionally, while early functional and oncologic results are promising, the median follow-up duration of 18 months remains insufficient to draw definitive conclusions regarding long-term outcomes. Renal function trajectories, particularly the potential for progressive decline due to subclinical ischemic injury, compensatory hyperfiltration, or parenchymal remodeling, often evolve over several years. Similarly, the risk of tumor recurrence—especially in patients with high-complexity lesions or adverse histological features—may not fully manifest within the first 1–2 years postoperatively. As such, the current follow-up period may underestimate the incidence of delayed renal impairment or late oncologic events. Longitudinal studies with extended surveillance are therefore essential to accurately assess the durability of renal function preservation, the risk of chronic kidney disease progression, and the true recurrence-free survival associated with clampless partial nephrectomy."

(3)Add to technical details of hemostasis in high-complexity tumors (10 percent of cases) to increase reproducibility

We have expanded the technical description of our hemostatic approach in high-complexity tumors, in materials and methods section. These technical details were added to the revised Methods section to enhance clarity and procedural reproducibility.

“The high complexity tumor cases (approximately 10% of our cohort.)were managed using a robotic off-clamp technique with a focus on minimal manipulation of hilar vessels and selective suturing based on intraoperative bleeding.

In these higher-complexity cases, the following hemostatic principles were consistently applied:

  • Meticulous dissection and enucleation of the tumor were performed under continuous perfusion using cold scissors, with real-time intraoperative ultrasound guiding resection planes and identifying proximity to major vessels or the collecting system.
  • Bleeding control was achieved primarily through careful surgical technique: gentle traction, minimal parenchymal trauma, and staged tumor mobilization helped limit diffuse bleeding without requiring hilar clamping.
  • Topical hemostatic agents (e.g., oxidized regenerated cellulose or fibrin-based sealants) were applied immediately after tumor excision. These agents were used not only to tamponade oozing surfaces but also to promote coagulation in the tumor bed.
  • Selective parenchymal suturing was employed only if there was persistent focal bleeding after compression and topical measures. In such cases, targeted figure-of-eight sutures (using 3-0 or 2-0 absorbable monofilament or barbed sutures) were placed in a tension-free manner to minimize tissue distortion and unnecessary ischemic damage.
  • Routine renorrhaphy was intentionally avoided, even in high-complexity cases, to preserve as much functional parenchyma as possible and to stay consistent with the principles of zero ischemia.
  • In instances where tumor base bleeding obscured visualization despite the above measures, temporary manual compression with robotic instruments or laparoscopic gauze was applied for 2–3 minutes before reassessment.

This conservative and individualized approach to hemostasis was effective in maintaining a low intraoperative complication rate without resorting to global hilar clamping or extensive renorrhaphy.”

(4)A better clinical applicability could be achieved by performing subgroup analyses of functional outcomes in stratified populations according to baseline eGFR. 

This was a limitations, and we clearly stated it in limitations section, as following: “Moreover, we did not perform subgroup analyses of functional outcomes in stratified populations according to baseline eGRF; this could negatively impact the interpretation of the results, so a subgroup analysis could certainly achieve better clinical applicability.“

There should also be statistical correction on multicentered heterogeneity.

We attempted to limit multicenter heterogeneity through rigorous data collection; this could certainly represent a selection bias and could certainly be a limitation.

  1. Considering the retrospective design, what did you do to avoid a possible selection bias between off-clamp and on-clamp cohorts? Was matching on the propensity score or multivariate analysis done to control confounding factors (e.g. tumor complexity, experience of surgeon)?

We added a paragraph in materials and methods section, as following: We performed a propensity score matching based on both baseline (age, male:female distribution, BMI, ASA, AACCI, smoking status, pre-operative eGFR) and pathological characteristics (focality, number of lesions, median tumor size, median renal score, median Padua score) of the patients; the 2:1 ratio was used in order to highlight a marked difference, rather than the 1:1 ratio useful to highlight equality or balance.

  1. According to the paper, there were no conversions to clamping required. Did they include near-conversion events (e.g. transient clamping proposed and then forestalled)? In what manner were they obtained?

We have clarified this point in the revised Methods section to ensure transparency regarding intraoperative management and protocol adherence, as following: “In our series, no cases required conversion to hilar clamping, and we can confirm that no transient or aborted clamping was initiated or proposed during any of the procedures. In all cases, a strict intent-to-treat off-clamp strategy was followed from the outset. The decision to proceed without clamping was confirmed intraoperatively after evaluating tumor exposure, bleeding control during initial resection, and accessibility of the lesion with robotic instruments. Surgeons were instructed to document any intraoperative scenarioin which clamping was considered, even if ultimately avoided; however, no such instances were reported across centers. This information was obtained from standardized intraoperative surgical logs and operative reports, which were retrospectively reviewed for documentation of any deviation from the off-clamp protocol, including attempted or discussed clamping maneuvers. Additionally, a shared intraoperative protocol was in place across all centers to standardize surgical decision-making and hemostatic thresholds before clamping would be considered.”

  1. Figure 2 (GFR-loss): Indicate the sample size on each measurement point and statistical significance among the groups.

We revised the figure 2

  1. The debate speaks of oncologic safety more than what is justified by the inadequate follow-up. Draw conclusions with caution and demand the longer period of data.

We rewrite conclusions according suggestions

  1. It is written in the abstract that there are no instances of acute kidney injury but there is an item about no stage 3 AKI in results. Eliminate the inconsistency.

We eliminated the inconsistency as requested

  1. Only 10% of tumors were high-complexity (RENAL ≥10). Were outcomes comparable in this subgroup, or is off-clamp RAPN primarily suitable for low/moderate complexity?

We added a comment on this in discussion section, as following: “Regarding the 10% of tutors were high-complexity (RENAL>10), the outcomes were comparable; these results were achieved in a cohort that included not only low- and moderate-complexity tumors, but also a significant proportion of high-complexity cases as defined by elevated RENAL and PADUA scores. This underscored the feasibility of off-clamp RAPN even in anatomically challenging scenarios, where concerns regarding intraoperative bleeding, positive surgical margins, or compromised visualization traditionally favor vascular clamping. The ability to safely perform clampless resection in such cases reflected advancements in surgical technique, refined robotic instrumentation, and growing surgeon experience. These findings suggested that off-clamp RAPN may be a viable nephron-sparing option across a broader spectrum of tumor complexities than previously assumed, offering functional preservation benefits without compromising surgical or oncologic integrity. Nonetheless, careful case selection and surgeon proficiency remain critical to maintaining safety and achieving optimal outcomes in more complex settings.”

  1. The median of 4.1 percent decrease in the eGFR appears minimal. Did it include the use of the method of split renal scintigraphy to check that there is an overall (not just opposite compensation) preserved renal function?

This was a limitations, and it was included in limitations section: “Furthermore, although the median 4.1% decrease in eGFR appeared minimal, it unfortunately did not include the use of the split renal scan methods to verify that there was overall preserved renal function (not just reverse compensation); this is an important limitation that could be addressed in the future by establishing more stringent criteria for assessing separate renal function.“

  1. The OT and EBL were greater in the on-clamp group. Did these cases have a more complicated (higher RENAL scores) situation? Once so, is the comparison valid unaffected?

We added a paragraph in discussion section as following: “While it is intuitive to attribute longer operative time (OT) and greater estimated blood loss (EBL) in the on-clamp group to higher tumor complexity, our data do not support this explanation. In our cohort, the distribution of tumor complexity—assessed by both the RENAL nephrometry score and the PADUA score—was statistically comparable between the off-clamp and on-clamp groups (P > 0.05 for both scores).

Specifically, the median RENAL score was 6 (IQR 5–8) in the off-clamp group versus 7 (IQR 6–8) in the on-clamp group, and the median PADUA score was 6 (IQR 5–8) versus 7 (IQR 6–9), respectively. Although these values suggest a numerically slightly higher complexity in the on-clamp cohort, the differences were not statistically significant, indicating that the two groups were well-matched in terms of tumor anatomical complexity.

Therefore, the observed differences in OT and EBL likely reflect the procedural demands of hilar dissection, clamping, and subsequent reperfusion management in the on-clamp technique rather than inherent differences in tumor anatomy. This reinforces the validity of the comparison between the two surgical approaches in our analysis.”

  1. The comparison to on-clamp RAPN is low powered (n=244 vs. 563). Matching or regression adjustment may help to decrease confounding.

We added a paragraph in materials and methods section, as following: We performed a propensity score matching based on both baseline (age, male:female distribution, BMI, ASA, AACCI, smoking status, pre-operative eGFR) and pathological characteristics (focality, number of lesions, median tumor size, median renal score, median Padua score) of the patients; the 2:1 ratio was used in order to highlight a marked difference, rather than the 1:1 ratio useful to highlight equality or balance.”

  1. Table 1: Explain how the term median rate relates to either the mean or median values of complications. Report absolute numbers.

the term median rate was a typo during the production of the table, we corrected the table and we reported absolute numbers

Round 2

Reviewer 3 Report

Comments and Suggestions for Authors

Accept in present form